Anti-methicillin-resistant Staphylococcus aureus and antibiofilm activity of new peptides produced by a Brevibacillus strain

Ogunsile Abiodun 1
Songnaka Nuttapon 1 2
Sawatdee Somchai 1 2
http://orcid.org/0000-0001-7777-8062 Lertcanawanichakul Monthon 3
http://orcid.org/0000-0003-4428-4140 Krobthong Sucheewin 4
Yingchutrakul Yodying 5
http://orcid.org/0000-0003-0846-0100 Uchiyama Jumpei 6
http://orcid.org/0000-0003-2280-6649 Atipairin Apichart 1 2 apichart.at@mail.wu.ac.th
1 School of Pharmacy, Walailak University , Nakhon Si Thammarat , Thailand
2 Drug and Cosmetic Excellence Center, Walailak University , Nakhon Si Thammarat , Thailand
3 School of Allied Health Sciences, Walailak University , Nakhon Si Thammarat , Thailand
4 Center of Excellence in Natural Products Chemistry (CENP), Department of Chemistry, Faculty of Science, Chulalongkorn University , Bangkok , Thailand
5 National Center for Genetic Engineering and Biotechnology, National Science and Technology Development Agency , Pathum Thani , Thailand
6 Department of Bacteriology, Graduate School of Medicine Dentistry and Pharmaceutical Sciences, Okayama University , Okayama , Japan
Puia Zothan
Electronic publication date: 2023 Oct 2
Publication date: 2023
Volume: 11
Electronic Location ID: e16143
Received 2023 May 8; Accepted 2023 Aug 29
Copyright: © 2023 Ogunsile et al.
Copyright year: 2023
Copyright holder: Ogunsile et al.
License: This is an open access article distributed under the terms of the Creative Commons Attribution License, which permits unrestricted use, distribution, reproduction and adaptation in any medium and for any purpose provided that it is properly attributed. For attribution, the original author(s), title, publication source (PeerJ) and either DOI or URL of the article must be cited.
License URL: https://creativecommons.org/licenses/by/4.0/

Keywords: Antimicrobial peptide, Anti-MRSA peptide, Antibiofilm activity, Brevibacillus spp., SPR19

Funding: Walailak University 2565),/2565 Walailak University Graduate Research Fund CGS-RF-2022/04 Walailak University Master Degree Excellence Scholarships ME11/2021 Walailak University Scholarships for High Potential Candidates to Enroll in Master Programs in Drug and Cosmetic Innovation 2/2021 This research was funded by Walailak University (Contract No. WUBG-014/2565), the Walailak University Graduate Research Fund (Contract No. CGS-RF-2022/04), Walailak University Master Degree Excellence Scholarships (Contract No. ME11/2021), and Scholarships for High Potential Candidates to Enroll in Master Programs in Drug and Cosmetic Innovation, Walailak University (Contract No. 2/2021). The funders had no role in study design, data collection and analysis, decision to publish, or preparation of the manuscript.

==============================
Background

Methicillin-resistant Staphylococcus aureus (MRSA) is listed as a highly prioritized pathogen by the World Health Organization (WHO) to search for effective antimicrobial agents. Previously, we isolated a soil Brevibacillus sp. strain SPR19 from a botanical garden, which showed anti-MRSA activity. However, the active substances were still unknown.

Methods

The cell-free supernatant of this bacterium was subjected to salt precipitation, cation exchange, and reversed-phase chromatography. The antimicrobial activity of pure substances was determined by broth microdilution assay. The peptide sequences and secondary structures were characterized by tandem mass spectroscopy and circular dichroism (CD), respectively. The most active anti-MRSA peptide underwent a stability study, and its mechanism was determined through scanning electron microscopy, cell permeability assay, time-killing kinetics, and biofilm inhibition and eradication. Hemolysis was used to evaluate the peptide toxicity.

Results

The pure substances (BrSPR19-P1 to BrSPR19-P5) were identified as new peptides. Their minimum inhibition concentration (MIC) and minimum bactericidal concentration (MBC) against S. aureus and MRSA isolates ranged from 2.00 to 32.00 and 2.00 to 64.00 µg/mL, respectively. The sequence analysis of anti-MRSA peptides revealed a length ranging from 12 to 16 residues accompanied by an amphipathic structure. The physicochemical properties of peptides were predicted such as pI (4.25 to 10.18), net charge at pH 7.4 (−3 to +4), and hydrophobicity (0.12 to 0.96). The CD spectra revealed that all peptides in the water mainly contained random coil structures. The increased proportion of α-helix structure was observed in P2−P5 when incubated with SDS. P2 (NH2-MFLVVKVLKYVV-COOH) showed the highest antimicrobial activity and high stability under stressed conditions such as temperatures up to 100 °C, solution of pH 3 to 10, and proteolytic enzymes. P2 disrupted the cell membrane and caused bacteriolysis, in which its action was dependent on the incubation time and peptide concentration. Antibiofilm activity of P2 was determined by which the half-maximal inhibition of biofilm formation was observed at 2.92 and 4.84 µg/mL for S. aureus TISTR 517 and MRSA isolate 2468, respectively. Biofilm eradication of tested pathogens was found at the P2 concentration of 128 µg/mL. Furthermore, P2 hemolytic activity was less than 10% at concentrations up to 64 µg/mL, which reflected the hemolysis index thresholds of 32.

Conclusion

Five novel anti-MRSA peptides were identified from SPR19. P2 was the most active peptide and was demonstrated to cause membrane disruption and cell lysis. The P2 activity was dependent on the peptide concentration and exposure time. This peptide had antibiofilm activity against tested pathogens and was compatible with human erythrocytes, supporting its potential use as an anti-MRSA agent in this post-antibiotic era.

Introduction

Staphylococcus aureus is a Gram-positive bacterium commonly present on the skin of healthy people. When the skin is injured, this pathogen can penetrate the skin or enters mucous membranes and bloodstream to cause serious illnesses such as wound infections, meningitis, endocarditis, pneumonia, and osteomyelitis (Nandhini et al., 2022). Formerly, penicillins were useful antibiotic drugs to treat such infections, but the cells developed drug resistance, resulting in severe health problems (Guo et al., 2020). Methicillin-resistant Staphylococcus aureus (MRSA) is an important strain that is resistant to methicillin and other commonly used β-lactam such as oxacillin, penicillin, and amoxicillin (Liu et al., 2022). The spread of MRSA is varied according to the geographic location in which its proportion of more than 20% was reported in Southeast Asia, European, and American regions (Prestinaci, Pezzotti & Pantosti, 2015). Currently, the World Health Organization (WHO) considers MRSA a high-prioritized pathogen based on the prevalence of antibiotic-resistant bacteria and the incidence of morbidity and mortality worldwide (Tacconelli et al., 2018). It encourages researchers globally to search for new substances to combat drug-resistant pathogens.

The mec A gene on the staphylococcal cassette chromosome (SCC) is responsible for drug resistance in MRSA strains. This inhibits the activities of β-lactam antibiotics by lowering the binding affinity to penicillin-binding protein 2a, which has the transpeptidase function to allow cell wall biosynthesis (Craft et al., 2019). Biofilm formation by S. aureus is also another cause of antimicrobial resistance. Biofilm is an encased community of microbial cells that allows the attachment of planktonic bacteria to biotic and abiotic surfaces (Flemming & Wuertz, 2019). The bacteria start to proliferate and form an irreversible attachment or mature biofilm. The biofilm is protected by an extracellular polymeric matrix layer primarily composed of oligosaccharides, DNA, and proteins, making pathogenic bacteria highly impenetrable during drug treatment that contribute to the virulence and chronic pathogenicity of the microorganism. MRSA has become a difficult-to-treat pathogen because of its ability to form a biofilm by reducing its susceptibility to antibiotic treatment by up to 1,000-folds. It requires a higher concentration of antimicrobial agents to eliminate biofilms than planktonic bacteria (Silva et al., 2021).

Soil bacteria are major sources of antimicrobial substances, which cover over 70% of antibiotic products (Cycoń, Mrozik & Piotrowska-Seget, 2019). Streptomyces and Bacillus are important genera because they produce many antibiotics such as erythromycin, streptomycin, vancomycin, and bacitracin (Hug et al., 2018). Brevibacillus is also known to be a useful bacterial genus. They are rod shape, endospore-forming, aerobic or facultative anaerobes, and Gram-positive or Gram-variable bacteria widely distributed across different habitats (Ray, Patel & Amin, 2020). Brevibacillus spp. are bacteria that are known to produce bioactive compounds with broad-spectrum therapeutic abilities. Some species, e.g., Brevibacillus laterosporus, have been widely used in agriculture as bioagent control (anti-insect and anti-nematode) (Hamze & Ruiu, 2022). These groups of bacteria have also been used to produce some natural products such as bacteriocin, brevibacillin, and brevibacillin V for peptide antibiotics and food preservation (Yang & Yousef, 2018; Wu et al., 2019). Brevibacillus spp. also produces some antimicrobial peptides (AMPs) with strong potency against multidrug-resistant pathogens such as MRSA and vancomycin-resistant Enterococcus faecalis (Yang et al., 2016; Clements-Decker et al., 2022). Our previous study isolated Brevibacillus sp. strain SPR19 (referred to as SPR19) from a botanical garden soil (Suan PRuekttas) in Nakhon Si Thammarat, Thailand. The results showed that SPR19 produced some bioactive substances against S. aureus and MRSA clinical strains (Songnaka et al., 2022b). However, these substances were unknown. The aim of this study was to purify and characterize these active substances. We also evaluated their antimicrobial activity, stability, mode of action, antibiofilm, and toxicity.

Materials and Methods

Bacterial cultivation

This study was approved by the Institutional Biosafety Committee (WU-IBC-65-003; 26 January 2022). Brevibacillus sp. SPR19 (GenBank accession number: MZ298491) was a soil bacterium isolated from a botanical garden in Nakhon Si Thammarat, Thailand, and deposited at Thailand Institute of Science and Technological Research (TISTR), Pathum Thani, Thailand with the collection number of TISTR 10170 (Songnaka et al., 2022b). It was cultured on Mueller-Hinton (MH) agar (Titan Biotech Ltd., Rajasthan, India) and incubated at 30 °C for 24 h. Tested bacteria (S. aureus TISTR 517 and MRSA isolate 142, 1096, and 2468 from Maharaj Nakhon Si Thammarat Hospital, Nakhon Si Thammarat, Thailand) were streaked on MH agar and incubated at 37 °C for 24 h (Memmert GmbH+ Co., Schwabach, Germany).

Purification of anti-MRSA substances

A colony of SPR19 was suspended in 0.9% NaCl solution, and its optical density (OD) at 625 nm was adjusted to 0.1. A total of 4 mL of cell suspension was added into 196 mL of Luria-Bertani (LB) broth (Titan Biotech Ltd., Rajasthan, India) before shaking at 30 °C, 150 rpm for 24 h. The cell-free supernatant (CFS) was prepared by centrifugation of cell culture at 10,000 × g and 4 °C for 15 min. The active substances in CFS were separated by ammonium sulfate precipitation, cation exchange chromatography (CIEX), and reversed-phase chromatography (RPC) as detailed in our previous study (Songnaka et al., 2022a). The purified samples were evaporated by a refrigerated speed vacuum concentrator. The dried substance was dissolved in deionized water and prepared in a two-fold serial dilution. A total of 100 μL of sample solutions was used to determine the antimicrobial activity by agar well diffusion method, using MRSA isolate 2468 as a tested bacterium. The purification balance sheet was recorded by measuring the dried weight of the anti-MRSA substance and calculating its antimicrobial activity in each step of purification by using the following equation:

Antimicrobial activity (arbitrary unit; AU) = 2n×1,000V where n is the reciprocal of the highest two-fold dilution, exhibiting a clear zone, and V is the sample volume in µL (Yang et al., 2018).

Evaluation of anti-MRSA activity

Agar well diffusion method

A cell suspension of tested strains (S. aureus TISTR 517 and MRSA isolate 142, 1096, and 2468) was prepared by adjusting the OD at 625 nm to 0.1 using a spectrophotometer (Thermo Fisher Scientific Inc., Waltham, MA, USA) and then swabbed on a sterile MH agar. Wells of 9 mm were punched on the solid media surface by a sterile tip, and 100 µL of purified substance was loaded before incubating the plates at 37 °C for 24 h. Vancomycin (30 µg) (Sigma-Aldrich Co., St. Louis, MO, USA) was used as a control drug, whereas fresh medium was used as a blank control. The result was collected from three independent experiments in which the diameter of the zone of inhibition was measured by a vernier caliper and reported as mean ± SD (Lei et al., 2020). The significant difference was determined by Student’s t-test at a p-value of less than 0.05.

Broth microdilution assay

The assay was conducted in accordance with the guideline of the Clinical and Laboratory Standard Institute (CLSI). The cell suspension of tested strains (S. aureus TISTR 517 and MRSA isolate 142, 1096, and 2468) was prepared by adjusting the turbidity at the OD 625 nm to 0.1 (1 × 108 CFU/mL). The cell suspension was prepared in a 20-fold dilution using Cation-Adjusted Mueller-Hinton broth (CAHMB) as a diluent. A total of 10 µL of the diluted bacterial preparation and 100 µL of pure substances (0.25–64 µg/mL) were loaded into the 96-well plates. The plates were incubated at 37 °C for 24 h. Control wells were performed for broth (CAMHB) and inoculum. Vancomycin was tested as a control drug. The experiment was performed in triplicate for each tested strain. Minimum inhibition concentration (MIC) was described as the lowest concentration of pure substances that showed no visible bacterial growth. A total of 10 µL from the samples in the above wells was spread on the MH agar and incubated at 37 °C for 24 h to measure the minimum bactericidal concentration (MBC). MBC was the lowest concentration of the substances that showed no colony on MH agar after 24 h incubation (CLSI, 2018).

Peptide sequencing

The purified peptides were dissolved in 0.1% formic acid and 1% acetonitrile (ACN) and then subjected to a high-resolution liquid chromatography-tandem mass spectrometer (LC-MS/MS) (Thermo Fisher Scientific Inc., Waltham, MA, USA). A total of 3 µL of the samples was loaded on a C18 column. The LC elution condition and MS/MS operational parameters were adopted from our previous study (Songnaka et al., 2022a). Peak Studio X (Bioinformatics Solutions Inc., Waterloo, CA, USA) was used for peptide identification (Krobthong & Yingchutrakul, 2020). The acceptable de novo peptide sequences were obtained by scoring the sequences with average local confidence (ALC) above 80%. The mass spectrometry proteomics data have been deposited to the ProteomeXchange Consortium via the PRIDE partner repository (Perez-Riverol et al., 2022) with the dataset identifier PXD042035. The mass, pI, net charge, and hydrophobicity of the peptides were estimated by HeliQuest (Gautier et al., 2008).

Circular dichroism (CD) spectropolarimetry

The peptide solutions (0.60 mg/mL) were prepared by deionized water or 50 mM SDS. The circular dichroism (CD) spectra were recorded by a J-815 spectropolarimeter (JASCO Corporation, Tokyo, Japan) at 25 °C in the wavelength between 190 and 260 nm. The samples were put in a 1 mm cuvette cell, and the parameter settings were as follows; a bandwidth of 1.0 nm, a scanning speed of 100 nm/min, and a data interval of 0.1 nm. The corresponding diluent was used to subtract its signal from the sample spectrum. The experiment was performed in triplicates, and the estimation of the peptide secondary structure was analyzed by the CONTINLL program (Sreerama & Woody, 2000).

Polyacrylamide gel electrophoresis and gel overlay assay

The purified peptides (5 μg) were analyzed on a 15% sodium dodecyl sulfate-polyacrylamide gel (SDS-PAGE). The gel was soaked in a Tris-glycine buffer pH 8.3 and ran at a constant voltage of 100 V. The protein bands in a one-half gel were detected by silver staining, and a protein marker was also loaded in the gel for comparison of band position. Another half gel was immersed in a fixing solution that contained 5% acetic acid and 25% methanol for 30 min and washed with distilled water for 3 h. The gel was poured with 18 mL of soft MH agar containing MRSA isolate 2468 to determine the inhibition zone after incubation at 37 °C for 24 h (Songnaka et al., 2022a).

Stability tests for the active peptide

The stability of the anti-MRSA peptide in response to temperature, pH, and proteolytic enzyme was determined by incubating the sample solutions (80 µg/mL) in the following conditions. Heat treatment was conducted at 25, 50, 75, and 100 °C for 1 h. The sensitivity to acid and basic solutions was performed at different pH of 3.0, 4.0, 5.0, 6.0, 7.0, 8.0, 9.0, and 10.0 for 2 h at 37 °C. The pH was subsequently adjusted to 7.0 before further evaluation. Sensitivity to proteolysis was tested by incubating the purified peptides with trypsin, α-chymotrypsin, and proteinase K (1.0 mg/mL) (Vivantis Technologies Sdn. Bhd., Selangor Darul Ehsan, Malaysia) at 37 °C for 2 h. The antimicrobial activity of all treated samples was determined by agar well diffusion against S. aureus TISTR 517 and MRSA isolate 2468. The residual activity was calculated as a percentage based on the proportion of the inhibition zone of treated and untreated samples. The untreated peptides and fresh medium were used as the negative and blank controls, respectively (Baindara et al., 2016). Each experiment was conducted in triplicates, and the data were presented as mean ± SD. The result analysis was done by Student’s t-test, and the p-value of less than 0.05 accounted for statistical significance.

Scanning electron microscopy (SEM)

The cell suspension of S. aureus TISTR 517 and MRSA isolate 2468 was prepared in 0.9% NaCl, and adjusted the OD at 625 nm to 0.1. The purified peptide (1× MIC) was introduced into the cells and incubated at 37 °C for 1, 3, and 16 h. The untreated cells were used as a negative control. Furthermore, the cells were obtained by centrifugation at 10,000× g and 4 °C for 5 min and gently put on the surface of a glass slide. The cell-fixing was done by adding 2.5% glutaraldehyde in 0.1 M phosphate buffer at pH 7.2 and incubating at 4 °C for 24 h. Cells on slides were later washed with 0.1 M phosphate buffer pH 7.2, and 1% osmium tetroxide (OsO4) in distilled water was added for 1 h as the post-fixation. The dehydration process employed a stepwise gradient with increasing ethanol concentrations (20–100%) for 15 min in each step. A critical point dryer (Quorum Technologies Ltd., Lewes, UK) was used to dry the fixed cells on the slides and covered them with gold (Cressington Scientific Instrument Ltd., Watford, UK). The cells were captured under 50,000× magnification of a scanning electron microscope (Songnaka et al., 2022a).

Time-killing kinetic assay

The overnight culture of S. aureus TISTR 517 and MRSA isolate 2468 were prepared with CAMHB and standardized at OD 625 nm to 0.1. Various peptide concentrations (2×, 10×, and 20× MIC) were also prepared. A total of 50 µL of different peptide concentrations was added to 50 µL of diluted cell suspension (20-fold dilution). The mixture was incubated at 37 °C for 0, 1, 2, 3, 6, 12, 18, and 24 h, respectively. At each point, 10 µL of the incubated mixture was used to prepare a 10-fold serial dilution using CAMHB as a diluent. Furthermore, 10 µL from the serially diluted sample was plated on a MH agar and incubated at 37 °C for 24 h. Cell colonies were counted, and the result was reported as colony-forming units (CFU)/mL. Experiments were done in triplicates, and the significant difference (p-value < 0.05, 0.01, and 0.001) were determined by one-way ANOVA for multiple comparisons between the treated and untreated samples (Yasir, Dutta & Willcox, 2019).

Cell membrane permeability assay

The sytox green uptake was used to assess the integrity of the cell membrane of S. aureus TISTR 517 and MRSA isolate 2468. Phosphate buffered saline (PBS) supplemented with 0.2% CAMHB (PBS-CAMHB) was used to prepare a bacterial suspension from a 16 h culture. The cell culture was centrifuged at 10,000× g and 4 °C for 15 min, and then the pellets were washed three times with PBS-CAMHB. The cell turbidity was adjusted at OD 625 nm equal to 0.1, and 100 µL of the standardized cell was added into a black 96-well microplate. The cells were mixed with the sytox green solution (5 μM) (Thermo Fisher Scientific Inc., Waltham, MA, USA) and incubated in a dark place for 15 min. A total of 100 µL of peptide solution were mixed to obtain the final concentrations between 2 and 128 µg/mL. The fluorescence was quantified at excitation and emission wavelength of 504 and 523 nm, respectively. Triton X-100 (1%) (AppliChem GmbH, Darmstadt, Germany) and non-treated samples were used as the positive and negative controls, while PBS-CAMHB only was used as a blank control (Yasir, Dutta & Willcox, 2019). Three independent experiments were performed. The statistically significant differences (p-value < 0.05) were evaluated by two-way ANOVA, using post hoc Tukey’s test for multiple comparisons between treatments and non-treatments.

Inhibition of biofilm formation

The overnight culture of S. aureus TISTR 517 and MRSA isolate 2468 was prepared in soybean-casein digest (SCD) broth. The cells were standardized using the same medium to achieve 1 × 108 CFU/mL. The cell suspension was mixed with peptide at various concentrations (0.5–256 µg/mL) in a 96-well plate. The plates were then incubated at 37 °C for 48 h. Then, the planktonic bacteria cells were removed from the plates by washing them with PBS. Thereafter, 100 µL of methanol was added to the plate to fix the biofilm-associated cells. The plates were dried and stained with 0.1% crystal violet for 15 min before rinsing twice with PBS. A total of 100 μL of 0.33% acetic acid was finally added to resolubilize the stained crystal violet. The plates were incubated at 37 °C for 30 min, and the supernatants were transferred to a new 96-well plate. The absorbance at 570 nm was measured by a microplate reader (Thermo Fisher Scientific Inc., Waltham, MA, USA) to determine the biofilm formation. Vancomycin and untreated samples were used as the positive and negative controls, whereas SCD media without cells was a blank control (Garrison et al., 2015; Zhu et al., 2020). One-way ANOVA was used to analyze the significant differences among treated samples compared to the untreated controls at various p-values. Dose-response curves of P2 and vancomycin on biofilm inhibition were constructed using GraphPad Prism software version 9.5.1 for Windows (GraphPad Software, Boston, MA, USA). The data (logarithm of the peptide concentration and biofilm formation) were fitted by a nonlinear regression with four parameter logistic, and the half maximal inhibitory concentrations (IC50) with their 95% confidence interval (CI) were calculated.

Biofilm eradication assay

A single colony of S. aureus TISTR 517 and MRSA isolate 2468 was inoculated into SCD broth and incubated at 37 °C overnight. The bacteria were standardized to achieve 1 × 108 CFU/mL, and 100 µL of cell suspension was transferred to a 96-well microplate. The plates were incubated at 37 °C for 48 h for biofilm formation. The free bacteria cells were removed from the plates by washing them with PBS. Peptide solutions at the concentrations (0.5–256 µg/mL) were added to the microplate and incubated at 37 °C for 24 h. Subsequently, PBS was used to rewash the plate, and 100 µL of PBS was loaded into each well. The plate was sonicated for 30 min at room temperature, and the samples were taken and made a ten-fold serial dilution. A total of 10 µL of the diluted samples was plated on MH agar for colony count. Vancomycin and untreated samples were used as the positive and negative controls. Only SCD media without bacteria was used as the blank (Yuan et al., 2019). The significant difference was evaluated by one-way ANOVA (p-value < 0.05) compared to untreated controls.

Hemolytic assay

The Ethics Committee in Human Research at Walailak University has approved this study (WUEC-22-242-01; 15 August 2022). The written consent was obtained from a subject involved in the research. Whole blood (5 mL) was collected from a human volunteer and centrifuged at 700× g for 8 min. The pellets were washed 3 times with PBS before diluting to obtain 0.5% erythrocyte suspension. A total of 50 µL of erythrocytes was added to a 96-well plate, and the purified peptide in PBS (50 μL) was mixed to achieve the final peptide concentration of 1, 2, 4, 8, 16, 32, 64, 128, 256, and 512 µg/mL. The mixtures were then incubated at 37 °C for 1 h. They were further centrifuged at 1,000× g for 10 min, and the supernatant was loaded into a microplate before measuring the absorbance at 540 nm. The positive and negative controls were 1% Triton X-100 and untreated samples, respectively, while PBS was used as a blank control. This assay was carried out in three independent experiments. The percent hemolysis was calculated by the following equation:

%Hemolysis=[Asam−Ablank][Apos−Ablank]×100

where Asam is the sample absorbance, Apos and Ablank are the absorbance of positive and blank controls, respectively (Oddo & Hansen, 2017). The therapeutic index (TI) of hemolysis was measured as a ratio between the highest peptide concentration that exhibited 10% hemolysis and its geometric mean MIC value (Sengkhui, Klubthawee & Aunpad, 2023).

Results

Isolation of anti-MRSA substances

The CFS of SPR19 was pooled, and the anti-MRSA substances were purified by a three-stepwise method; salt-precipitation, CIEX, and RPC consecutively. There was no inhibition zone against MRSA isolate 2468 found from the precipitate after ammonium sulfate precipitation at a 25% saturation level. The substances after salt precipitation at 50% saturation exhibited the inhibition zone. It was shown that the active substances were purified 2.13-fold with a percent yield of 80.00 in this step (Table 1). The active fractions were subjected to CIEX, by which the partially purified substances had a purification fold of 4.26 and a percent yield of 28.00. The final step of RPC could fractionate five anti-MRSA substances (BrSPR19-P1 to BrSPR19-P5; referred to henceforth as P1 to P5) (Fig. 1) with a purification fold between 4.81 and 7.92, and a percent recovery in the range from 2.00 and 7.20.

Table 1 The purification balance sheet of anti-MRSA substances from SPR19.

The antimicrobial activity was measured by agar well diffusion assay against MRSA isolate 2468.

Sample	Volume (mL)	Total weight (mg)	Total activity (AU)	Specific activity (AU/mg)	Purification fold	Yield (%)	
Cell-free supernatant	1,000	740.00	40,000	54.05	1.00	100.00	
Salt precipitation	50	278.25	32,000	115.00	2.13	80.00	
CIEX	70	48.65	11,200	230.22	4.26	28.00	
RPC (BrSPR19-P1)	10	3.01	800	265.78	4.92	2.00	
RPC (BrSPR19-P2)	24	4.49	1,920	428.04	7.92	4.80	
RPC (BrSPR19-P3)	36	7.02	2,880	410.26	7.59	7.20	
RPC (BrSPR19-P4)	42	3.23	840	259.74	4.81	2.10	
RPC (BrSPR19-P5)	31	6.53	2,480	379.87	7.03	6.20	
Note:

CIEX, cation exchange chromatography; RPC, reversed-phase chromatography.

Figure 1 Reversed-phase chromatogram of anti-MRSA substances from SPR19.

The peak signal was detected at 214 nm. Lines indicate the fractions with anti-MRSA activity.

Antimicrobial activity by microdilution assay

The pure substances (P1–P5) from RPC were collected to evaluate for the MIC and MBC values against S. aureus TISTR 517 and MRSA isolates. P2 had the highest inhibitory activity with a MIC of 2 µg/mL against all tested bacteria and was followed by P3 (4 µg/mL), P4 and P5 (8 µg/mL), and P1 (32 µg/mL) (Table 2). Furthermore, P2 had the highest killing activity against all tested bacteria with MBC between 4 and 8 µg/mL and was followed by P3 and P4 (16 µg/mL), P5 (16–32 µg/mL), and P1 (64 µg/mL). Vancomycin had MIC and MBC values of 2 µg/mL against all tested strains. P2 was the most active substance from SPR19 as it had the lowest MIC and MBC values. Its inhibitory activity was the same as vancomycin, although its bactericidal activity differed among tested pathogens. Then, P2 was selected for further investigation. The result from the agar well diffusion assay found that the P2 activity was dependent on its concentration. P2 at the 1× MIC level did not exhibit inhibition zones in all tested bacteria, but the larger inhibition zones were significantly observed when incubated with increasing P2 concentrations (5×, 10×, 20×, and 40× MIC) (Table 3; Fig. S1). Vancomycin (30 μg or 150× MIC) showed a significantly large inhibition zone from 20.73 ± 0.23 to 23.90 ± 0.10 mm. It is evident that P2 could serve as an anti-MRSA agent and should be further studied for stability, mode of action, and toxicity.

Table 2 Broth microdilution test of active substances from SPR19.

The MIC and MBC of pure substances from RPC were determined against S. aureus TISTR 517 and MRSA isolate 142, 1096, and 2468. Vancomycin was used as a control drug. The experiment was performed in triplicate (n = 3).

Substance	S. aureus
TISTR 517	MRSA
isolate 142	MRSA
isolate 1096	MRSA
isolate 2468	
MIC (µg/mL)	MBC (µg/mL)	MIC (µg/mL)	MBC (µg/mL)	MIC (µg/mL)	MBC (µg/mL)	MIC (µg/mL)	MBC (µg/mL)	
BrSPR19-P1	32.00	64.00	32.00	64.00	32.00	64.00	32.00	64.00	
BrSPR19-P2	2.00	4.00	2.00	8.00	2.00	8.00	2.00	4.00	
BrSPR19-P3	4.00	16.00	4.00	16.00	4.00	16.00	4.00	16.00	
BrSPR19-P4	8.00	16.00	8.00	16.00	8.00	16.00	8.00	16.00	
BrSPR19-P5	8.00	16.00	8.00	32.00	8.00	32.00	8.00	32.00	
Vancomycin	2.00	2.00	2.00	2.00	2.00	2.00	2.00	2.00	

Table 3 Antimicrobial susceptibility testing of P2 by agar well diffusion assay.

The antimicrobial activity of the P2 substance was determined by measuring the inhibition zones of the tested bacteria. Vancomycin was used as a control drug. The experiment was performed in triplicate, and the results are presented as mean ± SD.

Sample	S. aureus
TISTR 517
(mm)	MRSA
isolate 142
(mm)	MRSA
isolate 1096
(mm)	MRSA
isolate 2468
(mm)	
1× MIC P2 (0.2 μg)	0.00 ± 0.00	0.00 ± 0.00	0.00 ± 0.00	0.00 ± 0.00	
5× MIC P2 (1 μg)	9.80 ± 0.61a	10.52 ± 1.31a	11.17 ± 0.15a	11.40 ± 0.17a	
10× MIC P2 (2 μg)	12.40 ± 0.53b	12.56 ± 3.01	14.87 ± 0.31b	14.67 ± 0.72b	
20× MIC P2 (4 μg)	15.03 ± 0.45c	15.97 ± 1.55	17.77 ± 0.57c	17.70 ± 0.95c	
40× MIC P2 (8 μg)	17.70 ± 0.20d	19.20 ± 0.30d	19.70 ± 0.17d	20.30 ± 1.25d	
Vancomycin (30 μg)	20.73 ± 0.23e	23.47 ± 0.59e	23.10 ± 0.36e	23.90 ± 0.10e	
Note:

The significant difference was analyzed by Student’s t-test (p-value < 0.05) compared to a1×, b5×, c10×, d20×, and e40× MIC of P2.

Characterization of anti-MRSA peptides

Five anti-MRSA substances from RPC were subjected to a LC-MS/MS to fragment the daughter ions of the b- and y-ion series (Table S1). A de novo algorithm derived the peptide sequences based on the mass difference between two fragment ions and the confidence scores of each amino acid (Fig. 2). The result revealed that the anti-MRSA peptides from SPR19 had a length between 12 and 16 residues. The presence of non-polar amino acids (valine, leucine, and phenylalanine) contributed to peptide hydrophobicity. The positively (lysine) and negatively (aspartate and glutamate) charged residues dominated the hydrophilic structures and charges of the peptides. The predicted physiochemical properties of these peptides were as follows; mass (1,437.89 to 1,837.01 Da), isoelectric point (pI 4.25 to 10.18), net charge at pH 7.4 (−3 to +4), and hydrophobicity (0.12 to 0.96) (Table 4). The most occurring amino acid in P1 was glutamic acid (25.00%), whereas valine was abundant in P2 (41.67%), P3 (35.71%), P4 (23.08%), and P5 (23.08%). This result reflects that all peptides are amphiphilic, which suggests potential anti-MRSA activities against tested pathogens.

Figure 2 Identification of anti-MRSA peptides from SPR19 by de novo sequencing.

A tandem mass spectrometer fragmented the b- and y-ions of five peptides (P1–P5), and the corresponding mass spectra were shown in (A–E), respectively. The peptide sequences were achieved from the mass spectra with the local confidence scores of each amino acid.

Table 4 Predicted properties of anti-MRSA peptides from SPR19.

The peptide sequences were calculated for mass, pI, net charge, and hydrophobicity (Gautier et al., 2008).

Sample	Peptide sequence	Length
(residues)	Mass (Da)	pI	Net charge at pH 7.4	Hydrophobicity	
BrSPR19-P1	TLEEGKSVEFEVVDQK	16	1,837.01	4.25	−3	0.12	
BrSPR19-P2	MFLVVKVLKYVV	12	1,437.89	9.70	+2	0.96	
BrSPR19-P3	LTKKLVVKVLYKVV	14	1,630.13	10.18	+4	0.60	
BrSPR19-P4	FAELLWVVVADKT	13	1,490.76	4.37	−1	0.74	
BrSPR19-P5	FAELLGEVVVWKT	13	1,490.76	4.53	−1	0.72	

Determination of peptide secondary structure

The circular dichroism (CD) spectra of all peptides were measured by a spectropolarimeter. The result revealed that all peptides (P1-P5) in the water had similar CD spectra with a negative band between 190 and 230 nm (Fig. 3A). The random coil (48.5–50.9%) conformations were predicted to be the main secondary structure (Table 5). SDS is generally used to mimic the negatively charged and hydrophobic part of bacterial membranes (Krenev et al., 2020; Wu et al., 2020). When SDS was incubated with the peptides, only P2–P5 had spectral changes with a positive band between 190 and 205 nm and a negative band between 205 and 240 nm. The great magnitude at 195 and 220 nm was observed in the positive and negative bands, respectively (Fig. 3B). It was estimated that the proportion of α-helix structure was increased (2.5–5.1%) and that of turn conformation was decreased (5.8–8.2%) in the mixture of peptides (P2–P5) and SDS (Table 5). The structural change of P2–P5 might result in higher anti-MRSA activity when compared to that of P1. In addition, the CD spectra of P1 did not change in the presence of SDS, implying that the P1 had an alternative action on antimicrobial activity.

Figure 3 CD spectra of active peptides (P1–P5) from SPR19.

The peptides (0.6 mg/mL) were dissolved in (A) deionized water and (B) 50 mM SDS solution. The spectra were determined at 25 °C in the wavelength between 190 and 260 nm.

Table 5 The secondary structure content of anti-MRSA peptides from SPR19.

The CD spectra of peptides (P1–P5) in the different solvents (deionized water and 50 mM SDS) were predicted the secondary structures by the CONTINLL program (Sreerama & Woody, 2000).

Sample	Solvents	α-Helix	β-Strand	Turn	Random coil	
BrSPR19-P1	H2O	0.2	27.5	21.5	50.8	
50 mM SDS	0.4	26.5	19.0	54.1	
BrSPR19-P2	H2O	0.3	30.7	20.0	49.0	
50 mM SDS	5.0	34.4	6.1	54.5	
BrSPR19-P3	H2O	0.5	28.5	20.1	50.9	
50 mM SDS	15.1	30.3	5.8	48.8	
BrSPR19-P4	H2O	0.3	31.1	20.1	48.5	
50 mM SDS	3.9	33.6	7.9	54.6	
BrSPR19-P5	H2O	0.3	27.7	21.3	50.7	
50 mM SDS	2.5	34.5	8.2	54.8	

SDS-PAGE and gel overlay assay of peptide

The 15% SDS-PAGE gel showed the peptide bands of P1–P5 with an estimated size between 5 and 10 kDa (Fig. S2A). After the gel was overlaid with soft agar containing MRSA isolate 2468, it displayed an inhibition zone at the same location of peptides on the stained gel (Fig. S2B). It can be deduced that all purified peptides exhibited anti-MRSA activity.

Stability test of the purified peptide (P2)

P2 was the most active peptide from SPR19. It was further used to determine its stability. The antimicrobial activity of P2 against the tested bacteria (S. aureus TISTR 517 and MRSA 2468) was constant, although the temperature was raised to 100 °C for 1 h (p-value > 0.05). The remaining activity was between 98.41 ± 0.55% and 100.90 ± 1.35%, indicating that this peptide was thermostable (Table 6). The addition of proteolytic enzymes (α-chymotrypsin, trypsin, and proteinase K) to the peptide unaffected the P2 function by which the residual activity was 97.54 ± 0.48% to 100.46 ± 1.37% (p-value > 0.05). It suggested that P2 was stable to proteolytic enzymes. However, the P2 activity against S. aureus TISTR 517 significantly decreased when incubated under a pH less than 5 (p-value < 0.05), but it was stable at a pH between 5 and 10. There was a fluctuation in the antimicrobial activity for MRSA isolate 2468 at pH between 3 and 10, but it was not significantly different (p-value > 0.05). Collectively, P2 activity against tested bacteria was tolerant in the pH range from 3 to 10 with the activity of more than 90%.

Table 6 Stability studies of P2 under various conditions (temperature, proteolytic enzyme, and pH).

The residual activity of P2 was determined by measuring the diameter of the inhibition zone between treated and untreated samples. Each experiment was performed in triplicate, and the result was mean ± SD.

Conditions	% Remaining activity	
S. aureus TISTR 517	MRSA isolate 2468	
Temperature (°C)			
Untreated sample	100.00 ± 0.00	100.00 ± 0.00	
25	100.90 ± 1.35	99.19 ± 1.74	
50	100.55 ± 1.42	100.71 ± 3.25	
75	98.41 ± 0.55	100.02 ± 1.71	
100	99.66 ± 1.09	98.97 ± 2.78	
Proteolytic enzyme			
Untreated sample	100.00 ± 0.00	100.00 ± 0.00	
Sample + α-chymotrypsin	98.52 ± 1.01	98.02 ± 1.84	
Sample + trypsin	97.57 ± 4.34	98.96 ± 2.68	
Sample + proteinase-K	97.54 ± 0.48	100.46 ± 1.37	
pH			
Untreated sample	100.00 ± 0.00	100.00 ± 0.00	
pH 3	92.62 ± 1.37*	98.19 ± 0.28	
pH 4	92.25 ± 1.44*	96.94 ± 2.66	
pH 5	97.78 ± 1.93	96.95 ± 5.77	
pH 6	101.29 ± 1.39	99.10 ± 0.82	
pH 7	99.81 ± 3.55	101.48 ± 3.04	
pH 8	95.76 ± 1.94	101.82 ± 0.34	
pH 9	97.05 ± 2.10	101.29 ± 2.07	
pH 10	94.66 ± 4.46	99.82 ± 1.14	
Note:

* Indicate the significant difference by Student’s t-test at p-value < 0.05 compared to untreated samples.

Effect of the anti-MRSA peptide on the bacterial cells and membrane

The SEM micrograph revealed that both S. aureus TISTR 517 and MRSA isolate 2468 in the untreated condition appeared to be unbroken, firm, and round cells with smooth surfaces throughout the study period (Figs. 4A and 4B). When incubated with P2 at 1× MIC value for 1 h, these tested cells had a deflated and irregular shape. The cells were more damaged under a longer duration of incubation (3 and 16 h), by which the cell surface was rough and deformed, resulting in cell collapse and lysis. In addition, the complex formations between sytox green dye (a fluorescent dye) and nucleic acids generally imply that the cellular membrane permeability after being treated with AMPs. Before the addition of the tested substances, the fluorescence intensity showed no significant difference compared to the non-treatment conditions. The overall intensities for P2 (1× to 64× MIC) and Triton X-100 (1%) in both bacteria were 0.95 ± 0.19 to 1.18 ± 0.14 and 1.05 ± 0.06 to 1.17 ± 0.12 folds, respectively, in comparison to the non-treatment conditions. There was an increase in fluorescence immediately after incubation with P2. At the end of the study (1,440 min or 24 h), the intensity in S. aureus TISTR 517 increased by 1.36 ± 0.02 to 5.80 ± 0.76 folds when treated with P2 at concentrations ranging from 1× to 64× MIC (Fig. 5A). Similarly, an increased fluorescence was observed in MRSA isolate 2468 by which the intensity was changed from 1.53 ± 0.14 to 9.11 ± 0.22 folds when incubated with P2 at concentrations between 1× and 64× MIC (Fig. 5B). The intensity rapidly increased within 30 and 150 min after introducing P2 (4× to 64× MIC) in S. aureus TISTR 517 and MRSA isolate 2468, respectively, indicating that membrane disruption occurred during this time. However, the intensity remained constant thereafter, suggesting the complete binding of dye and nucleic acid. Furthermore, the incubation of sytox green dye and P2 peptide (64× MIC) did not result in an increase in fluorescence signal (Figs. 5A and 5B). The higher intensity was found in MRSA isolate 2468, implying that P2 could alter membrane permeability in MRSA isolate 2468 more than S. aureus TISTR 517. Interestingly, the fluorescence intensity was significantly increased when the tested strains were treated with P2 at concentrations of more than 2× MIC (p-value < 0.05). The result showed that the uptake of sytox green dye was dependent on the peptide concentration. Triton X-100 (1%) was used as a positive control, and the fluorescence intensity was elevated from 1.17 ± 0.12 to 16.31 ± 1.54 and 1.05 ± 0.06 to 17.49 ± 0.36 folds for S. aureus TISTR 517 and MRSA isolate 2468, respectively, when compared to the non-treatment. It was demonstrated that the fluorescence of the Triton X-100 treated sample exhibited a time-dependent pattern. This intensity was unchanged significantly since 960 min or 16 h. Furthermore, the fluorescence level in the treatment of MRSA isolate 2468 with Triton X-100 was higher than that of S. aureus TISTR 517, suggesting this resistant bacterium was more sensitive to membrane disruption.

Figure 4 SEM micrograph of bacterial cells after treatment with anti-MRSA peptide (P2).

(A) S. aureus TISTR 517 and (B) MRSA isolate 2468 were treated with P2 at the 1× MIC level for 1, 3, and 16 h and compared to the non-treatment conditions.

Figure 5 Membrane permeability was altered by anti-MRSA peptide (P2).

The cell cultures of (A) S. aureus TISTR 517 and (B) MRSA isolate 2468 were incubated with various concentrations of P2 (1×, 2×, 4×, 8×, 16×, 32×, and 64× MIC levels). The cells with 1% Triton X-100 were used as a positive control, whereas the non-treated sample was a negative control. The cell-free samples (P2 at 64× MIC with sytox green dye) were also measured. The fluorescence intensity was determined at 532 nm between 0 and 1,440 min. The inset was the expanded view of the boxed region. Each experiment was performed in triplicate. The significant difference in fluorescence intensity was determined between treated and non-treated samples (p-value < 0.05).

Bacterial killing kinetics of anti-MRSA peptide

The bacterial growth in the absence of P2 increased between 3 and 18 h of the incubation period. It reached the stationary phase after 18 h with an increased growth of 2.64 and 2.80 log10 CFU/mL at 24 h for S. aureus TISTR 517 and MRSA isolate 2468, respectively, when compared with the initial inoculum (Figs. 6A and 6B). The growing rate constant and the killing half-time were calculated based on the linear regression between log10 of viable cells and time. After the addition of P2 to S. aureus TISTR 517 for 3 h, the growth rates were 0.04 (95% CI [0.01–0.07]), −0.06 (95% CI [−0.01 to −0.12]), and −0.13 (95% CI [−0.07 to −0.18]) h−1 for 1×, 5×, and 10× MIC, respectively (Fig. 6A). The decrease in growth rates was observed between 3 and 12 h in which the rates were −0.04 (95% CI [−0.09 to 0.00]), −0.42 (95% CI [−0.34 to −0.49]), and −0.52 (95% CI [−0.44 to −0.60]) h−1, and the calculated killing half-times were 75.82 (95% CI [59.50–101.79]), 10.68 (95% CI [8.18–13.90]), and 9.84 h (95% CI [6.62–14.41]) for 1×, 5×, and 10× MIC, respectively. Similarly, the growth rates of MRSA isolate 2468 in the first 3 h were 0.13 (95% CI [0.09–0.18]), −0.04 (95% CI [−0.13 to 0.05]), and −0.12 (95% CI [−0.01 to −0.22]) h−1 after incubation with P2 at 1×, 5×, and 10× MIC, respectively (Fig. 6B). The rates between 3 and 12 h were 0.03 (95% CI [0.00–0.06]), −0.31 (95% CI [−0.24 to −0.38]), and −0.52 (95% CI [−0.45 to −0.60]) h−1, and the predicted killing half-times were 82.22 (95% CI [51.39–172.18]), 11.61 (95% CI [9.35–14.40]), and 9.73 h (95% CI [6.52–14.33]) when treating with P2 at 1×, 5×, and 10× MIC levels, respectively. P2 at all concentrations significantly inhibited cell growth during 6–24 h compared to the non-treatment at the corresponding time (p-value < 0.05). P2 at 5× and 10× MIC showed complete bacterial killing at 18 and 12 h of incubation, respectively. It indicates that P2 activity depends on the incubation time and peptide concentration.

Figure 6 Killing kinetics of P2 against (A) S. aureus TISTR 517 and (B) MRSA isolate 2468.

P2 at concentrations of 1×, 5×, and 10× MIC values was incubated with tested bacteria at specified intervals. The viable cells were counted by diluting the samples and spreading them on MH agar. The samples without P2 were used as the non-treatment. The result was presented as a log10 of the viable cell (CFU/mL). Significant differences (p-value < 0.05) were assessed by one-way ANOVA to compare the treated and non-treated samples at the corresponding time.

Biofilm inhibition and eradication

The biofilm formation of tested bacteria can be inhibited by AMPs, which could be measured by a crystal violet assay in a 96-well microplate. The result showed that the biofilm of S. aureus TISTR 517 was significantly decreased (72.82 ± 8.91% and 74.78 ± 2.64%) when the cells were pre-incubated with P2 and vancomycin, respectively, at 2 μg/mL or 1× MIC value (p-value < 0.05) (Fig. 7A). Interestingly, higher concentrations of P2 and vancomycin (4 μg/mL or 2× MIC level) significantly inhibited biofilm formation of MRSA isolate 2468 (64.59 ± 4.90% and 68.25 ± 4.21%, respectively) (p-value < 0.01), suggesting that the biofilm-forming capacity was an important virulence factor for this resistant strain (Fig. 7B). The biofilm production of both bacteria was significantly decreased, although it did not totally diminish when the concentration of both tested substances increased (8-256 μg/mL) (p-value < 0.01 or < 0.001). The Hill equation was used to quantify the half-maximal inhibitory concentrations (IC50) of substances on antibiofilm activity. The IC50 of P2 and vancomycin was 2.92 (95% CI [2.45–3.49]) and 4.84 (95% CI [4.08–5.74]) µg/mL, respectively for S. aureus TISTR 517 (Figs. S3A and S3B). The higher concentration of P2 and vancomycin at 4.84 (95% CI [4.26–5.51]) and 5.91 (95% CI [5.20–6.70]) µg/mL, respectively, were used for half-maximal inhibition of biofilms from MRSA isolate 2468 (Figs. S3C and S3D). It indicated that P2 had more potency than vancomycin in biofilm inhibition. This could be explained by the result that P2 killed planktonic cells more than vancomycin before the survived persistent cells formed biofilm. In addition, the biofilm of the resistant strain was more persistent than the sensitive bacteria. Treatment of preformed S. aureus TISTR 517 and MRSA isolate 2468 biofilms with P2 revealed that the cell reduction was a function of peptide concentration (Figs. 8A and 8B). P2 at the concentration of 4 µg/mL significantly decreased both mature biofilms compared to untreated samples (p-value < 0.05), whereas it completely killed these persistent cells at 128 µg/mL. Similarly, vancomycin at 64 µg/mL significantly reduced the persistent cells in the biofilms of MRSA isolate 2468, and the drug at 128 µg/mL entirely eradicated both biofilms. Although P2 could significantly reduce the number of persistent cells at a lower concentration than vancomycin, it completely eradicated these cells at the same vancomycin concentration. This might result from the high concentration of peptide or drug (128 μg/mL) that killed almost all cells in the biofilm environment. It indicated that P2 and vancomycin required a high concentration (64× MIC) to destroy mature biofilms, while those substances at low concentrations (2× and 1× MIC, respectively) killed planktonic cells of S. aureus TISTR 517 and MRSA isolate 2468.

Figure 7 Inhibition of biofilm formation by P2 and vancomycin against (A) S.aureus TISTR 517 and (B) MRSA isolate 2468.

The cells were incubated with tested substances at 37 °C for 48 h. The biofilm production was assessed by staining with 0.1% crystal violet. The biofilm formation is expressed as a percentage by dividing the absorbance of treated samples by that of untreated samples. Each result was obtained from three independent experiments and presented as mean ± SD. The significant difference was analyzed by one-way ANOVA compared to the untreated controls at p-value < 0.05 (*), 0.01 (**), and 0.001 (***), respectively.

Figure 8 Biofilm eradication assay of P2 and vancomycin.

Biofilms of (A) S. aureus TISTR 517 and (B) MRSA isolate 2468 were formed in the 96-well microplates for 48 h. Different concentrations of P2 and vancomycin were incubated with biofilms for 24 h at 37 °C. An aliquot of the sample was diluted and spread on MH agar. The viable cells were counted, and the results were presented as a mean of log10 viable cells ± SD. Each measurement was performed in three independent experiments. An asterisk (*) indicates the statistical difference analyzed by one-way ANOVA using a p-value at 0.05 compared to untreated samples.

In vitro hemolytic assay

Hemolysis is used to monitor the toxicity of AMPs on the mammalian cell membrane. Generally, the hemolytic activity of less than 10% was safe for human use (Amin & Dannenfelser, 2006). In this study, the P2 peptide had a percentage of hemolysis below 10% (0.84 ± 1.42% to 9.92 ± 1.04%) in concentrations between 1 and 64 µg/mL. In contrast, the increased activity (16.62 ± 0.85% to 72.03 ± 5.61%) was found when incubated with higher concentrations (128–512 µg/mL) (Fig. 9). Although the degree of hemolysis depended on the P2 concentration, it had a low hemolytic effect at the inhibitory concentrations. The index threshold of P2 was estimated to be 32, indicating that the peptide was highly selective towards bacterial cells compared to erythrocyte cells.

Figure 9 Hemolytic activity of P2 on human erythrocytes.

P2 was prepared in concentrations between 1 and 512 µg/mL and incubated with erythrocytes for 1 h. The release of hemoglobin is presented as % hemolysis by measuring the absorbance at 540 nm compared to the positive control (1% Triton X-100).

Discussion

Bacterial resistance is a top health problem globally. The effectiveness of various antimicrobial drugs has been undermined by these resistant pathogens. It was estimated that 4.95 million people were associated with the deaths from antibacterial resistance in 2019 (Antimicrobial Resistance Collaborators, 2022). MRSA caused more than 100,000 deaths in 2019, and its prevalence was different across countries as the resistance rates varied from 20% to 68% in Asia, America, and Africa (Lee et al., 2018). MRSA is on the target list of WHO to investigate potential natural products that can replace the declining conventional drugs to manage drug-resistant pathogens (Tacconelli et al., 2018). In this study, we continued to screen soil bacterial Brevibacillus sp. SPR19 for anti-MSRA agents. SPR19 was identified by 16S rRNA sequencing, and the phylogenetic tree analysis showed that this strain had high similarity (97.95%) with Brevibacillus halotolerans strain LAM0312 (Songnaka et al., 2022b).

AMPs are secondary metabolites composed of short chains of amino acids (usually less than 50). They form an integral part of the innate immune system of living organisms, acting as the first-line barrier or defense system that protects against microbial invasions (Hassan et al., 2023). Brevibacillus spp. are one of the bacteria to produce new AMPs for therapeutic possibilities. Brevibacillus brevis was credited for the invention of gramicidin S for treating Gram-positive and Gram-negative bacterial and fungi infections (Li et al., 2020). It was also reported that Brevibacillus laterosporus produced brevibacillin against foodborne pathogenic bacteria and phytopathogenic fungi (Wu et al., 2019). AMPs have emerged as potential alternatives to conventional antibiotics (Zainal Baharin et al., 2021). Generally, AMPs have a net positive charge (+2 to +11), allowing an electrostatic interaction with negatively charged phospholipids in the bacterial membranes. They are amphipathic molecules, which make them soluble and incorporate into the lipid bilayer of the cell membrane, leading to membrane perforation, cell lysis, and death (Castillo-Juárez et al., 2022).

In the present study, five anti-MRSA peptides were separated from SPR19 by RPC. P1 was first eluted from the column because it had the lowest hydrophobicity and high polarity because of four negatively charged amino acids (glutamate). This was supported by a previous study that determined the peptide retention in RPC when using acetonitrile as an eluting solution and trifluoroacetic acid (TFA) as an ion-pairing agent. They used the synthesized peptides with close molecular weight but different hydrophobicity as the model peptides. The uniform distribution of peptide peaks in the chromatogram was observed in which the retention time was directly proportional to their hydrophobicity (Krokhin & Spicer, 2009). P2 and P3 were subsequently eluted because they had higher hydrophobicity with net charges of +2 and +4, respectively. The peptide with a more positive charge could delay the elution order because it was highly masked by an ion-pairing agent (TFA) that increased the overall hydrophobicity. It was demonstrated that increasing the net charge of peptides (+1, +3, and +5) increased retention time (Shibue, Mant & Hodges, 2005). P4 and P5 contained several non-polar amino acids (valine and leucine) to give high hydrophobic properties and had a negative net charge because of aspartate and glutamate. The peptide hydrophobicity was predicted to play a key role in their late elution. Although P4 and P5 had the same molecular weight and net charge, P5 had a higher pI than P4. Under the acidic condition by TFA in the mobile phase, the elution of P5 was slower than that of P4. It was consistent with a previous investigation, showing that DAEFGHDSG-NH2 (pI 4.02) and DAEFRHDSGY-NH2 (pI 4.54) had RPC retention times of 3.75 and 7.24 min, respectively, under the isocratic elution by 0.1% TFA and ACN (Baczek, Walijewski & Kaliszan, 2008). Therefore, the separation in RPC is influenced by the sequence and net charge of the peptide, hydrophobicity of the peptide and ion-pairing agent, ACN concentration in the mobile phase, and alkyl chain of the stationary phase (Erckes & Steuer, 2022).

P1–P5 from SPR19 were potential anti-MRSA peptides because their MIC and MBC values against the tested bacteria ranged from 2 to 32 and 4 to 64 μg/mL, respectively. Vancomycin, as the positive control, showed inhibitory and bactericidal concentrations at 2 μg/mL. A recent systematic study classified AMPs based on the composition of amino acids. The content of one or more amino acids in the sequence with more than 25% was defined as the rich AMPs. Leucine-rich peptides were most abundant in the APD3 Antimicrobial Peptide Database, whereas valine-rich AMPs were predominantly produced in bacteria (Decker, Mechesso & Wang, 2022). Seven amino acids, including glutamate, were occasionally observed in the AMPs in this database. In this study, P1 and P2 were glutamate- and valine-rich peptides, respectively, whereas lysine and valine were rich in P3. Furthermore, P4 and P5 were close to the valine-rich peptides as their valine abundances were 23%. Therefore, the anti-MRSA activity of P1–P5 might result from the repetition of the same amino acids that caused the unique properties of these peptides (Decker, Mechesso & Wang, 2022). Interestingly, P2 (molecular mass 1,437.89 g/moL) had the highest potency among the isolated peptides and exhibited similar anti-MRSA activity on the non-adherent cells compared to vancomycin (molecular mass 1,449.20 g/moL) as the MIC values were 1.39 and 1.38 μM, respectively. In addition, the P2 activity was dose-dependent because the larger inhibition zones were observed when increasing the P2 concentration.

P1–P5 were identified as new AMPs because they were not recorded in the APD3 Antimicrobial Peptide Database (https://aps.unmc.edu/database; 3,578 peptides; accessed on 30 January 2023) and the Database of Antimicrobial Activity and Structure of Peptides (DBAASP) (https://dbaasp.org/search; 20,259 peptides; accessed on 30 January 2023) (Wang, Li & Wang, 2016; Pirtskhalava et al., 2021). Sequence similarities between P1–P5 and AMPs from bacteria listed in those databases were determined by pairwise alignments using MatGAT version 2.01 (Matrix Global Alignment Tool) with BLOSUM62 scoring matrix (Campanella, Bitincka & Smalley, 2003). The result showed that anti-MRSA peptides from SPR19 (P1–P5) exhibited the highest similarity (46–79%) to plantaricin A, tridecaptin A1, bogorol analogs, brevilaterin V4, AN5 analogs, and BrSPR20-P2 (Table 7). Interestingly, BrSPR19-P2 (NH2-MFLVVKVLKYVV-COOH) was similar to BrSPR20-P2 (NH2-VVMNLLVKVLKYVV-COOH) from our previous study, suggesting these peptides belonged to the same group of AMP in Brevibacillus genus and exhibited similar stability (Songnaka et al., 2022a).

Table 7 Pairwise alignments between P1–P5 from SPR19 and AMPs from APD3 and DBAASP databases.

Peptide names and sequences, their bacterial producers, and percent similarity were shown.

Peptide	APD3 database	Sequence	Sequence similarity (%)	DBAASP database	Sequence	Sequence
similarity (%)	
BrSPR19-P1	Tridecaptin A1
(Paenibacillus terrae NRRL B-30644)	VKGSWSKKFEVIA	50.00%	Plantaricin A
(Lactobacillus plantarum C11)	YSLQMGATAIKQVKKLFKKWGW	45.50%	
BrSPR19-P2	BrSPR20-P2
(Brevibacillus sp. SPR-20)	VVMNLLVKVLKYVV	78.60%	BrSPR20-P2
(Brevibacillus sp. SPR-20)	VVMNLLVKVLKYVV	78.60%	
BrSPR19-P3	Bogorol K
(Brevibacillus laterosporus MG64)	TVRIIVKVVKYLV	71.40%	BrSPR20-P2
(Brevibacillus sp. SPR-20)	VVMNLLVKVLKYVV	64.30%	
BrSPR19-P4	AN5-1
(Paenibacillus alvei AN5)	YSKSLPLSVLNP	53.80%	AN5-2
(Paenibacillus alvei AN5)	FCKSLPLPLSVK	53.80%	
BrSPR19-P5	Bogorol L
(Brevibacillus laterosporus MG64)	LVVLAVAVVVWLR	53.80%	Brevilaterin V4
(Brevibacillus laterosporus)	XVKVVVKVLKYLX	53.80%	

Generally, AMPs have secondary structures such as α-helix, β-strand, both α-helix and β-strand, and random coil structures (Hassan et al., 2023). They can bind to the cell surface of bacteria through the interaction between the amphiphilic part of the peptides and the acyl chains of the lipids. Conformational rearrangements of AMPs occur and then cause membrane leakage and disintegration, resulting in bacterial cell lysis (Luong, Thanh & Tran, 2020). The secondary structure of P1–P5 in water was mainly random coils. The presence of glycine, aspartate, and serine in the sequences was predicted to be coil-forming residues, whereas other amino acids (valine, isoleucine, threonine, and tryptophan) had the propensities for β-structure (Costantini, Colonna & Facchiano, 2006). Interestingly, the major conformations of all peptides were still coil structures in the SDS solution, but only P2–P5 obviously showed structural change with a more folded α-helix. The negatively charged and hydrophobic part of SDS, mimicking the bacterial cell membrane, influenced structural rearrangements of P2–P5 to exert anti-MRSA activity. Furthermore, P1 had a similar secondary structure in the water or SDS solution, reflecting an alternative mechanism for its action. It was postulated that the net negative charge of P1 could repel the negative charge of SDS. P1 also had the least hydrophobicity. Therefore, SDS ineffectively bound to P1 and barely caused the structural change. On the other hand, it was the unique characteristic of P1 to maintain its original structure. AMPs (arenicin, LL-37, and RR12) adopted a predominant β-strand or random coil structure in an aqueous solution, but they underwent a conformational change with an increase in α-helix when complexed with SDS (Dannehl, Travkova & Brezesinski, 2012; Krenev et al., 2020; Wu et al., 2020). In addition, some peptides (OM19R analogs) had the characteristics of random coil, and their structures did not significantly change in the presence of SDS, supporting that AMP-facilitated intracellular activities (increased DNA binding, reduced synthesis of intracellular proteins, and decreased intracellular ATP) influenced a microbial inhibition rather than membrane disintegration (Zhao et al., 2021).

P2 had the highest anti-MRSA activity and was studied in further experiments. It maintained strong stability with the residual activity of more than 90% against temperature (25–100 °C), pH (3–10), and proteolytic enzymes (α-chymotrypsin, trypsin, and proteinase K). The small peptide might easily renature to native structures of disordered and β-strand characters after heat treatment and contributed to antimicrobial activity, indicating high thermostability of P2. P2 resisted hydrolysis under acidic or basic conditions. It was also tolerant to proteolytic enzymes, although the specific cleavage sites were presented in the P2 sequence such as aromatic amino acids for proteinase K and α-chymotrypsin and basic amino acids for trypsin. It predicted that the high hydrophobicity of the peptide tended to form agglomerates or aggregates to decrease their surface contact with polar molecules and enzymes, resulting in reduced hydrolysis and enzymatic cleavage. It was supported by the reversible aggregates of therapeutic peptides or proteins such as insulin, forming relatively stable units towards degradation reactions than individual molecules (Brange & Langkjoer, 1993). However, the question that remains is whether the peptide is intact or undergoes cleavage while still maintaining a functional fragment under these conditions. Therefore, it is recommended to analyze the treated peptide using mass spectrometry in further study to address this limitation.

Anti-MRSA activity of P2 was mediated by pore formation, membrane disruption, and cell lysis based on SEM, membrane permeability, and killing-kinetic analyses. P2 at 1× MIC showed the inhibitory effect on tested pathogens during 24 h of the study. This peptide at higher concentrations (5× and 10× MIC) reduced cell growth in a time-dependent manner and exhibited bactericidal activity at 18 and 12 h, respectively, supporting that the killing rate depended on the P2 concentration. The bacterial membrane was immediately altered after P2 addition (4× to 64× MIC), in which the increased fluorescence from the complexes of impermeable dye and bacterial DNA occurred within 0.5 and 2.5 h in S. aureus TISTR 517 and MRSA isolate 2468, respectively. This could suggest that membrane permeation took place during this period by which P2 stimulated pore formation on the cell membrane, promoting the penetration of the dye into the microbial cells. The fluorescence intensity was constant after that time, suggesting the equilibrium between dye and bacterial chromosomes. However, this effect was not concentration-dependent between 1× and 2× MIC of P2 as no signal differences were significantly observed compared to non-treatment samples. The fluorescence of the positive control was higher than that of P2-treated cells at the end of the study, suggesting that P2 (1× to 64× MIC) was less toxic compared to 1% Triton X-100. Several previous studies showed consistent results in our investigations, demonstrating common mechanisms of AMPs to act as membranolytic agents (Yasir, Dutta & Willcox, 2019; Li et al., 2021). Melittin, melimine, Mel4, and synthesized peptides (WIK-10, WIR-10, and WIK-14) could accumulate on the cell surfaces due to interactions between peptides and bacterial membrane. These peptides permeabilized the cell membrane and rapidly allowed the sytox green dye to penetrate as a function of peptide concentration, suggesting their antimicrobial activities through destabilization of the cell membrane or membranolysis (Yasir, Dutta & Willcox, 2019; Park et al., 2022). Membranolytic AMPs (melittin, LL-37, and polymyxin B) generally bind to cell membranes by electrostatic interactions between positively charged AMPs and negatively charged microbial surfaces. The teichoic acids or lipopolysaccharides are the important targets of AMPs in the cell wall or outer membrane for Gram-positive or Gram-negative bacteria, respectively. Structural changes of the peptides occur upon binding to the cell membrane and cause potential membrane alterations through distinct mechanisms such as barrel-stave, carpet, and toroidal modes (Moretta et al., 2021; Li et al., 2021). Some AMPs (indolicidin, teixobactin, and temporin-L) can function via non-membranolytic mechanisms by translocating across the membrane and affecting normal cellular functions such as inhibition of nucleic acid, protein, or cell wall synthesis (Moretta et al., 2021). Based on our finding on membrane permeability and killing kinetics, we proposed that P2 (more than 4× MIC) induced rapid membrane permeability within 0.5 or 2.5 h in S. aureus TISTR 517 and MRSA isolate 2468, respectively. The peptide progressively inhibited bacterial growth from 2 to 12 h and ultimately resulted in complete cell death at 12 or 18 h. Although P2 is demonstrated to cause membrane disruption and cell lysis, its intracellular activity is interesting for further study to fully understand the mechanism of this peptide.

Biofilm is an important virulence mechanism in many bacterial pathogens. The cells irreversibly adhere to the biotic and abiotic surfaces and are surrounded by an extracellular matrix in response to stress conditions (Flemming & Wuertz, 2019). This type of bacteria significantly has different morphology, physicochemical properties, and antibiotic susceptibility, making biofilm more difficult to treat and eradicate than planktonic or free-floating bacteria. S. aureus and MRSA develop a biofilm to decrease or prevent the diffusion of antimicrobial drugs into the cells, resulting in more drug resistance (10–1,000 times) than their planktonic forms. It requires high doses and long-term treatment of antimicrobial drugs (Tuon et al., 2023). Our study revealed that P2 (IC50 2.92 and 4.84 μg/mL) was more potent than vancomycin (IC50 4.84 and 5.91 μg/mL) to inhibit biofilm formations of S. aureus TISTR 517 and MRSA isolate 2468, respectively on the plastic surface of 96-well plates. Interestingly, P2 and vancomycin eradicated these biofilms at a higher concentration (128 μg/mL) that was equivalent to 64× MIC. These results were consistent with other previous studies. Synthesized AMPs (D-Bac8, D-Omiganan, and D-WMR derivatives) had the same 1× MIC value (8 μg/mL) against planktonic S. aureus and MRSA strains (Zapotoczna et al., 2017). Complete biofilm inhibition of these tested peptides was observed at higher concentrations than 1× MIC (128, 256, and 256 μg/mL), whereas the eradication of mature biofilm was found at concentrations of 256, 1,024, and 1,024 μg/mL, respectively (Zapotoczna et al., 2017). Vancomycin was commonly used to treat S. aureus-associated biofilm, and its higher concentration (16.0–68.2 µg/mL) was required for half-maximal inhibition against statically grown biofilms of S. aureus and MRSA (Zapotoczna et al., 2017). The high concentration of vancomycin (1,000× to 4,000× MIC) was effectively found to eradicate biofilm of the susceptible and resistant strains of S. aureus, indicating that the response variability to antibiofilm activity was dependent on drug concentration, biofilm maturation, and exposure time (Chen et al., 2020). Combination therapy between vancomycin and other antibiotics (tigecycline and rifampin) effectively reduced S. aureus biofilms, emphasizing the potential need for combination therapy instead of monotherapy (Tuon et al., 2023). Therefore, the optimizing combination of P2 and vancomycin or other antibiotics should be included in future work to benefit the treatment of infections associated with S. aureus and MRSA biofilms.

Hemolysis is typically used as an initial toxicity assessment of AMPs by which the peptide amphipathicity and hydrophobicity contribute to cytotoxicity (Greco et al., 2020). Peptide hydrophobicity is the dominant factor that facilitates nonspecific and disruptive interactions with the zwitterionic mammalian membrane and buries its highly hydrophobic face into the membrane (Hollmann et al., 2016). Previous studies showed that some AMPs (innate defense regulator (IDR-1018) and ponericin W1) caused the lysis of erythrocytes in a concentration-dependent manner, and these analogs with decreased hydrophobicity could reduce their hemolytic activities or slightly impair the integrity of erythrocytes at their effective concentrations (He, Stone & Deber, 2021; Jiale et al., 2021). Our study revealed that P2 made improper hemolysis at a concentration of more than 64 μg/mL, but its calculated therapeutic index was 32. It indicated that P2 had higher selectivity toward microbial cells than mammalian cells. Taken together, P2 is a novel AMP with high antimicrobial activity, stability, and safety to human erythrocytes. It can inhibit and destroy biofilms, making the P2 peptide a promising anti-MRSA agent in this post-antibiotic era.

Conclusions

Brevibacillus sp. SPR19 produced active substances against S. aureus and MRSA strains. Five novel peptides (P1–P5) were identified and showed potential anti-MRSA activity. All peptides had amphiphilic characters, and only P2–P5 exhibited structural changes upon binding to the membrane-mimicking agent (SDS). Interestingly, P2 (NH2-MFLVVKVLKYVV-COOH) had the highest anti-MRSA activity, by which its action was dependent on peptide concentration and exposure time. P2 could disrupt the cell membrane and caused bacterial death, indicating its membranolytic function. This peptide was more stable towards temperature, pH, and proteolytic enzymes. In addition, P2 inhibited biofilm formation more than vancomycin and effectively eradicated mature biofilms of the tested pathogens. It had low hemolytic activity and showed high selectivity on bacterial cells. However, the chemical synthesis of P2, its stability under physiological conditions (e.g., serum and high salt), and mammalian cell cytotoxicity should be investigated in future work to support the antimicrobial activity and safety of this peptide. This study has successfully discovered potential anti-MRSA agents that can develop for treating this resistant pathogen.

Supplemental Information

Supplemental Information 1 The b-ion and y-ion fragmentations of each anti-MRSA peptide from Brevibacillus sp. SPR19 by tandem mass spectrometry.

(A–E) BrSPR19-P1 to BrSPR19-P5.

Click here for additional data file.

Supplemental Information 2 Agar well diffusion assay.

The P2 peptide at the different concentrations (1×, 5×, 10×, 20×, and 40× MIC) exhibited inhibition zones against (A) S. aureus TISTR 517 and (B–D) MRSA isolate 142, 1096, and 2468, respectively. Vancomycin (30 μg or 150′ MIC) was used as a control drug.

Click here for additional data file.

Supplemental Information 3 SDS-PAGE and agar overlay assay.

(A) Silver-stained SDS-PAGE gel of peptides from SPR19. Lane M, protein marker; Lanes 1−5, P1−P5, respectively (B) Agar overlay assay with MRSA isolate 2468. The corresponding gel was overlaid by soft agar, containing the tested bacteria. The plate was incubated at 37 °C for 24 h.

Click here for additional data file.

Supplemental Information 4 Dose-response curve of P2 and vancomycin on inhibition of biofilm formation.

(A and B) S. aureus TISTR 517 and (C and D) MRSA isolate 2468 were treated with P2 and vancomycin, respectively, and the biofilm production was measured. The Hill model was used to calculate the concentration of tested substances that inhibited biofilm formation by 50% (IC50).

Click here for additional data file.

Supplemental Information 5 Raw data of Table 3 and 6 and Figure 5–9.

Inhibition zone from agar well diffusion assay (Table 3) and stability test (Table 6). Fluorescence intensity from cell-permeability assay (Fig. 5). Cell number from time-killing kinetics assay (Fig. 6). Absorbance from inhibition of biofilm formation (Fig. 7). Cell number from biofilm eradication (Fig. 8). Absorbance from hemolysis (Fig. 9).

Click here for additional data file.

We acknowledge the Center of Scientific and Technological Equipment, Walailak University, for the instrumental support. Furthermore, we would like to thank Faculty of Pharmaceutical Sciences, Chulalongkorn University, for the generous support of the spectropolarimeter.

Additional Information and Declarations

Competing Interests

Author Contributions

Human Ethics

Ethics

DNA Deposition

Data Availability

The authors declare that they have no competing interests.

Abiodun Ogunsile conceived and designed the experiments, performed the experiments, analyzed the data, prepared figures and/or tables, authored or reviewed drafts of the article, and approved the final draft.

Nuttapon Songnaka conceived and designed the experiments, performed the experiments, analyzed the data, prepared figures and/or tables, authored or reviewed drafts of the article, and approved the final draft.

Somchai Sawatdee conceived and designed the experiments, analyzed the data, prepared figures and/or tables, authored or reviewed drafts of the article, and approved the final draft.

Monthon Lertcanawanichakul conceived and designed the experiments, analyzed the data, prepared figures and/or tables, authored or reviewed drafts of the article, and approved the final draft.

Sucheewin Krobthong performed the experiments, analyzed the data, prepared figures and/or tables, authored or reviewed drafts of the article, and approved the final draft.

Yodying Yingchutrakul performed the experiments, analyzed the data, prepared figures and/or tables, authored or reviewed drafts of the article, and approved the final draft.

Jumpei Uchiyama analyzed the data, prepared figures and/or tables, authored or reviewed drafts of the article, and approved the final draft.

Apichart Atipairin conceived and designed the experiments, performed the experiments, analyzed the data, prepared figures and/or tables, authored or reviewed drafts of the article, project administration, and approved the final draft.

The following information was supplied relating to ethical approvals (i.e., approving body and any reference numbers):

This study was approved by the Ethics Committee in Human Research, Walailak University.

The following information was supplied relating to ethical approvals (i.e., approving body and any reference numbers):

This study was approved by the Institutional Biosafety Committee, Walailak University.

The following information was supplied regarding the deposition of DNA sequences:

The peptide sequences identified by MS are available in Pride: PXD042035.

https://www.ebi.ac.uk/pride/archive/projects/PXD042035.

The following information was supplied regarding data availability:

The raw data are available in the Supplemental File.

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
