# Peer review of "Anti-methicillin-resistant Staphylococcus aureus and antibiofilm activity of new peptides produced by a Brevibacillus strain"

_PeerJ, doi:10.7717/peerj.16143_

## Round 0.1 · original submission · Minor Revisions

Authors need to address the queries and incorporate the suggestions of the reviewers in the manuscript, to be considered for publication.

Reviewer 1 ·

Basic reporting

The manuscript submitted by Ogunsile et al. identifies the active substances (small-peptides) from soil bacteria (Brevibacillus sp.) which result in anti-MRSA activity. The authors have presented a detailed work characterizing the active substance and its anti-MRSA potency. The paper is recommended for publication after the author addresses the queries listed below and has made the required changes in the manuscript.

Experimental design

no comment

Validity of the findings

1) In Figure 3A, CD spectra of 5 peptides are shown both in water and SDS solution. In the text, the authors suggest that peptides in water have a beta-sheet structure. However, based on the spectra, which have a low intensity for various secondary structures, peptides appear to be in unstructured coil form. I would like authors to provide some other evidence if they have any, which can establish the peptides in water to be in beta-sheet form.

2) The killing kinetics data shown in Figure 6, suggest that even 10X MIC of P2 doesn’t show much difference at least till 3 hours of incubation. Significant changes in cell viability are observed only after 6 hours of incubation.
Moreover, the membrane permeability data shown in Figure 5 of the manuscript, has been misinterpreted by the authors. If we look at the positive control (1% TritonX-100), the fluorescence shows time-dependent enhancement, in both S. aureus and MRSA bacteria. However, in the case of P2, a sudden jump in fluorescence is observed just after the P2 addition, and thereafter fluorescence remains the same through 180 min of measurement, at all concentrations of P2.

Therefore, the no increase in fluorescence till 180 minutes (3 hours) coincides with bacteria-killing kinetics, as suggested above. I would like authors to re-perform these experiments and collect fluorescence data till 12-18 hours after incubation. I propose a substantial increase in the fluorescence after 6 hours of incubation.



3) The initial increase in the fluorescence just after the addition of P2 could be due to some interactions of sytox green dye with the P2 protein. This could be verified by recording the fluorescence of syntax green in a P2 solution cell-free medium).

Reviewer 2 ·

Basic reporting

No Comment

Experimental design

No Comments

Validity of the findings

No Comments

Additional comments

In this study, the authors have identified and isolated five novel anti-MRSA peptides from Brevibacillus sp. strainSPR19. Among them, Peptide P2 was the most active peptide and demonstrated to cause bacterial membrane disruption and cell lysis. The peptide P2 activity was dependent on the peptide concentration. In addition, this selected peptide has antibiofilm activity and showed no cytotoxicity for human erythrocytes, that indicates the selectivity of the peptide towards the bacterial cells. Results from the experiments, suggesting its potential use as an anti-MRSA agent in this post-antibiotic era. The authors have done extensive study for this work and provided all the necessary data, and the manuscript is well written and structured. And I believe this manuscript can be within the scope of the journal. I recommend this manuscript can be accepted after minor correction. However, authors need to address the following concerns and make corrections in the manuscript, to be considered for the publication.


Minor Comments,
Page no 14, line no 319 – “The threshold index (TI) of hemolysis”. I believe its therapeutic index.

Page no 17, line no 432, log10 CFU/mL should be changed to log10 CFU/mL.

Page 18, line 463, 464 – change IC50 to IC50

Page 22, line 626 – “Several previous studies showed consistent results in our investigations”. Please provide references.
Page 22, line 627 – “Synthesized AMPs (D-Bac8, D-Omiganan, and D-WMR derivatives) had the same 1x MIC value (8 g/mL) against planktonic S. aureus and MRSA strains.” Please provide reference.
Regarding peptide stability: Its impressive that, peptide is active even after incubating with proteinase enzymes/different pH/ high temperature but the activity could be possible even the peptide is not intact. How do you make sure that the peptide is intact and not cleaved. Probably analyzing by mass-spec could help.

Figure 5 data implies that membrane permeability is concentration dependent not time dependent (page 17, line 424). Please change that.
Also, from my experience, 1% triton X-100 should result in maximum damage to the bacterial cell membrane immediately and thus maximum fluorescence in less than 2 minuets. but your data shows that the fluorescence increases with time, in which case the P2 is more toxic than 1% triton X-100!!!, it doesn’t add up. Therefore, I would suggest repeating this experiment.


The data from membrane permeability and killing kinetics are not corelating. Peptide P2 could almost maximum damage the membrane at 8X MIC within 10 minuets on the other hand, why it takes longer time (~12 h) to kill the bacterial cells (based on the killing kinetics)? Please explain.

Reviewer 3 ·

Basic reporting

No comment

Experimental design

No comments

Validity of the findings

yes.

Additional comments

In this article, the authors isolated a novel AMP from Brevibacillus sp. with potential antimicrobial activities. The authors further tested the activity of the isolated AMP against drug-resistant bacteria as well as determined its sequence and stability against temperature and acidity. This article needs a few minor revisions before it can be published.

1. L159, how the diameters for the zone of inhibitions were measured?
2. L224 antimicrobial activity instead of assay.
3. Provide real images for agar well diffusion assay.
4. L372, please provide a reference for using SDS to mimic bacterial membranes
5. L377-379, this statement is too strong. There is no proof yet that structural change in AMP results in cell lysis.

---

## Round 0.2 · accepted · Accept

The authors have addressed all the reviewers' comments and is ready for publication.

Reviewer 1 ·

Basic reporting

no comments

Experimental design

no comments

Validity of the findings

no comments

Additional comments

The authors have performed suggested experiments and made required changes in the manuscript. I recommend the article to be accepted for publication.

Reviewer 2 ·

Basic reporting

No Comment

Experimental design

No Comment

Validity of the findings

No Comment